# Differential Expression of microRNAs in Hypertrophied Myocardium and Their Relationship to Late Gadolinium Enhancement, Left Ventricular Hypertrophy and Remodeling in Hypertrophic Cardiomyopathy

**DOI:** 10.3390/diagnostics12081978

**Published:** 2022-08-16

**Authors:** Chen Zhang, Hongbo Zhang, Lei Zhao, Zhipeng Wei, Yongqiang Lai, Xiaohai Ma

**Affiliations:** 1Department of Interventional Diagnosis and Treatment, Beijing Anzhen Hospital, Capital Medical University, 2nd Anzhen Road, Chaoyang District, Beijing 100020, China; 2Department of Radiology, Beijing Anzhen Hospital, Capital Medical University, 2nd Anzhen Road, Chaoyang District, Beijing 100020, China; 3Department of Cardiac Surgery, Beijing Anzhen Hospital, Capital Medical University, 2nd Anzhen Road, Chaoyang District, Beijing 100020, China

**Keywords:** hypertrophic cardiomyopathy, microRNA, cardiovascular magnetic resonance, myocardial fibrosis, cardiac remodeling

## Abstract

**Background:** Differential expression has been found in a variety of circulating miRNAs in patients with hypertrophic cardiomyopathy (HCM). However, study on myocardial miRNAs is limited and a lot of miRNAs were not studied in previous studies. **Methods:** Twenty-one HCM patients and four patients who died from non-cardiovascular diseases were prospectively recruited for our study. A total of 26 myocardial tissues were collected, which were stored in liquid nitrogen immediately for miRNA detection using the Agilent Human miRNA Microarray Kit. All HCM patients underwent cardiovascular magnetic resonance (CMR) examination before surgery and cvi42 software was used to analyze cardiac function and myocardial fibrosis. **Results:** Compared with the control group, the expression of 22 miRNAs was found to be significantly increased in the HCM group, while 46 miRNAs were found to be significantly decreased in the HCM group. The expression levels of hsa-miR-3960 and hsa-miR-652-3p were significantly correlated with left ventricular mass index (r = 0.449 and 0.474, respectively). Meanwhile, Hsa-miR-642a-3p expression was positively correlated to the quantification of late gadolinium enhancement (r = 0.467). **Conclusion****s:** Our study found that 68 myocardial miRNAs were significantly increased or decreased in the HCM group. Myocardial miRNA levels could be used as potential biomarkers for LV hypertrophy, fibrosis and remodeling.

## 1. Introduction

As the most common monogenic cardiovascular disorder, hypertrophic cardiomyopathy (HCM) is associated with mutations in 11 or more genes [1]. MicroRNAs (miRNAs) are short RNA molecules that regulate the post-transcriptional silencing of target genes. A single miRNA can target hundreds of mRNAs and influence the expression of numerous genes [2]. A variety of miRNAs have been found as important regulators of multiple phases in cardiac development [3,4,5]. Previous studies reported that many miRNAs play roles in the pathogenic mechanisms of heart failure, such as hypertrophy, remodeling, hypoxia and apoptosis [6,7]. In the study by Roberta et al. [8], circulating miR-29a was found to be significantly up-regulated in HCM patients and correlated with fibrosis and left ventricular (LV) hypertrophy. However, circulating miRNAs did not only originate in the myocardial tissue but in other involved organs. Assessment of myocardial miRNAs could better reflect the miRNA expression in hypertrophied myocardium. Moreover, only dozens of miRNAs were analyzed in previous studies [8,9]; other miRNAs may also show differential expression in HCM patients.

Cardiovascular magnetic resonance (CMR) plays an important role in clinical work, which provides a mechanism to assess vascular or cardiac function, anatomy, tissue characteristics and perfusion in a highly reproducible manner [10,11,12,13]. CMR measurements of LV myocardial thickening, myocardial mass, or infarct size were confirmed with high reproducibility and low variance in repeated samples [14]. Late gadolinium enhancement (LGE) images derived from CMR could be used for identifying the location and extent of myocardial necrosis. The extent of LGE closely mirrors the distribution of myocyte necrosis at early periods [15], which may be used as a valuable tool to predict major adverse cardiac events (MACE) and cardiac mortality [16].

In this study, we mainly aimed to characterize the myocardial miRNA profile of HCM, and investigate miRNAs that showed differential expression in hypertrophied myocardium. Then, the correlation between myocardial miRNA levels and CMR variables was assessed to explore potential myocardial miRNA biomarkers of myocardial fibrosis, LV hypertrophy and remodeling.

## 2. Materials and Methods

### 2.1. Study Population

This study was approved by the Ethics Committee of Beijing Anzhen hospital and written informed consent was obtained from all the subjects. Twenty-one patients diagnosed with HCM were recruited for our study and all patients underwent transaortic extended septal myectomy in Beijing Anzhen hospital from November 2019 to December 2020. The diagnosis of HCM was based on echocardiographic or CMR demonstration of a hypertrophied but nondilated LV (with maximal wall thickness >15 mm at end diastolic) in the absence of any other systemic or cardiac disorder causing a similar grade of hypertrophy [17]. The indications for surgical myectomy included patients with obstructive HCM who remain severely symptomatic; symptomatic patients with obstructive HCM who have associated cardiac disease requiring surgical treatment and patients’ voluntary acceptance [17]. Four normal myocardium specimens were used as controls, which were obtained from patients who died from non-cardiovascular diseases in Beijing Anzhen hospital from November 2019 to December 2020. Their myocardial tissues were stored in liquid nitrogen immediately for miRNA detection. A total of 21 patients with HCM (8 men and 13 women) and 4 healthy controls (2 men and 2 women) were included in this study. Patient’s clinical characteristics were listed in Table 1.

### 2.2. CMR Image Acquisition

CMR images were acquired with a 32-channel surface phased array cardiac coil in two different pieces of equipment (Ingenia 3.0T, Philips Healthcare, Best, Netherlands; Discovery MR750 3.0T, GE Medical Systems, Milwaukee, WI, USA) following routine scan protocol. Cardiac cine images were collected using balanced steady-state free precession (bFISP) sequences with retrospective cardiac gating. Images covered the whole LV from the base to the apex and each cardiac cycle had 25 phases. Parameters of cine images for GE Discovery MR750 were: TR/TE = 3.6/1.4 ms, FA = 60°, FOV = 380 × 380 mm^2^; for Philips Ingenia: TR/TE = 3.0/1.52 ms, FA = 45°, FOV = 270 × 270 mm^2^. LGE images were collected using a breath-hold 2D phase-sensitive inversion-recovery (PSIR) segmented gradient echo sequence 10 minutes after the contrast agent (Gadopentetate dimeglumine, Bayer Healthcare) was intravenously administered at a dose of 0.2 mmol/kg body weight. Parameters of LGE images for GE Discovery MR750 were: TR/TE = 6.2/2.9 ms, FA = 25°, FOV = 380 × 380 mm^2^; for Philips Ingenia: TR/TE = 6.1/3.0 ms, FA = 25°, FOV = 350 × 350 mm^2^. All slice thicknesses were 5 mm for long axis images and 8 mm for short axis images with no interval between slice locations.

### 2.3. CMR Image Analysis

Endocardial and epicardial borders of LV from the base to the apex were automatically delineated and manually adjusted using commercial software (cvi42, version 5.11.2, Circle Cardiovascular Imaging Inc., Calgary, AB, Canada) in both end systolic and end diastolic phase. Then, the left ventricular end systolic volume (LVESV), left ventricular end diastolic volume (LVEDV), left ventricular mass (LVM) and LVEF were calculated by the software. In addition, we further standardized LVESV, LVEDV and LVM by body surface area (BSA). We also manually measured maximal LV end diastolic wall thickness (EDWT) by an experienced observer (L.Z., with 10 years of CMR experience). After epicardial and endocardial borders were manually delineated by an experienced observer (L.Z., with 10 years of CMR experience) in the LGE images, a visually normal appearing area of myocardium without hyperenhancement was manually selected as a normal myocardial region of interest. We defined LGE as myocardium 6 standard deviations (SD) above the mean signal intensity.

### 2.4. Microarray Infomation

The Agilent Human miRNA Microarray Kit, Release 21.0, 8 × 60K (DesignID:070156) experiment and data analysis of the 26 samples were conducted by LB Technology Co., Ltd. (Beijing, China). The microarray contains 2570 probes for mature miRNA.

### 2.5. Experiment

Total RNA was quantified by the NanoDrop ND-2000 (Thermo Scientific, Waltham, MA, USA) and the RNA integrity (RIN) was assessed using the Agilent Bioanalyzer 2100 (Agilent Technologies, Santa Clara, CA, USA). The sample labeling, microarray hybridization and washing were performed based on the manufacturer’s standard protocols. Briefly, total RNA was dephosphorylated, denatured and then labeled with Cyanine-3-CTP. After purification, the labeled RNAs were hybridized onto the microarray. After washing, the arrays were scanned with the Agilent Scanner G2505C (Agilent Technologies).

### 2.6. Experimental Data Analysis

Feature extraction software (version10.7.1.1, Agilent Technologies) was used to analyze array images to get raw data. Next, the raw data were normalized with the quantile algorithm. The probes detected with at least 75.0 percent in any group were chosen for further data analysis. Differentially expressed miRNAs were then identified through fold change and the *p*-value was calculated using a t-test. The threshold set for up- and down-regulated genes was a fold change ≥ 2.0 and a *p*-value ≤ 0.05. Target genes of differentially expressed miRNAs were the intersection predicted with 2 databases (miRDB, miRWalk). GO analysis and KEGG analysis were applied to determine the roles of these target genes. Hierarchical clustering was performed to show the distinguishable miRNAs expression pattern among samples. The miRNAs extraction and screening process are shown in Figure 1.

### 2.7. Statistical Analysis

Statistical analyses were undertaken with the SPSS software (IBM SPSS Statistics for Windows, Version 23.0; IBM Corp., Armonk, NY, USA) and R programming language (version 3.4.2, Available online: http://www.r-project.org (accessed on 19 January 2022). Quantitative data were expressed as the median and interquartile range (IQR), and categorical variables were presented as frequencies or percentages. Spearman correlation analyses were used to evaluate the potential correlation between miRNA levels and CMR variables. The Mann–Whitney U test was used to compare the difference in quantitative data in two different groups, and the chi-square test or Fisher’s exact test were used to compare the difference of categorical variables between two different groups.

## 3. Results

### 3.1. Patients’ CMR and Echocardiography Findings

Patients’ CMR and echocardiography findings are listed in Table 2, including LVEF, LVEDVi, LVESVi, EDWT, quantification of LGE, LVMi, LV outflow tract gradient (LVOTO) and left atrial volume index (LAVi).

### 3.2. Up-Regulated and Down-Regulated miRNAs in HCM Patients

Compared with the control group, the expression of 22 miRNAs was found significantly increased in the HCM group, while 46 miRNAs were found significantly decreased in the HCM group (Table 3). The expression of myocardial miR-208b-3p, -221-3p, -224-3p were extremely up-regulated (with foldchange >10) and miR-218-5p, -4741, -5787, -208a-3p, -551b-3p, -4788, -575, -4466, -1246, -7150, -204-5p, -208a-5p, -6850-5p, -7847-3p were extremely down-regulated (with foldchange < −10).

### 3.3. Correlation between miRNA Levels and CMR Variables

The expression levels of hsa-miR-3960 and hsa-miR-652-3p were significantly and positively correlated with LVMi (r = 0.449, *p* = 0.036; r = 0.474, *p* = 0.026, respectively). The expression levels of hsa-miR-3679-5p and hsa-miR-7107-5p were significantly and positively correlated with LVEF (r = 0.486, *p* = 0.022; r = 0.454, *p* = 0.034, respectively), while the expression level of hsa-miR-499a-5p was significantly and negatively correlated with LVEF (r = −0.571, *p* = 0.005). Meanwhile, hsa-miR-642a-3p expression was positively correlated to the quantification of LGE (r = 0.467, *p* = 0.028). Then, hsa-miR-3141 and hsa-miR-3679-5p expression were found to be negatively correlated to LVESVi (r = −0.556, *p* = 0.007; r = 0.459, *p* = 0.032, respectively) (Figure 2 and Figure 3).

## 4. Discussion

We evaluated the expression of 2570 miRNAs in each myocardial specimen and found that the expression of 22 miRNAs was significantly increased and 46 miRNAs were significantly decreased in the HCM group. Then, seven miRNAs were found to be significantly correlated to CMR parameters.

We evaluated the expression of myocardial miRNA levels rather than circulating miRNA levels, so the result in this study is different from previous studies [8,9]. In the study by Roberta et al. [8], the expression of circulating miR-27a, -199a-5p, -26a, -145, -133a, -143, -199a-3p, -126-3p, -29a, -155, -30a, and -21 were found to be increased in HCM patients. In the study by Derda [9] et al., circulating expression of miR-155 was significantly decreased in both obstructive and non-obstructive HCM patients. However, in this study, the expression of these miRNAs was not significantly up-regulated or down-regulated in HCM patients. This result means that the expression of miRNAs was different between myocardial and circulation.

In the preliminary study on the expression of myocardial miRNAs, Kuster [18] et al. found that the expression of miRNA-10b was down-regulated and miRNA-204, -497, -184, -222 and -34 were up-regulated. In the study by Song [19] et al., expression of miRNA-451 was significantly down-regulated, and overexpression of miR-451 in neonatal rat cardiomyocytes could reduce cell size. In the study by Huang [20] et al., the expression of miR-221, miR-222 and miR-433 was significantly up-regulated. The results in our study were partly similar to these studies as we also found up-regulated expression of miR-221 and down-regulated expression of miR-451, but no similar results were found in other miRNAs.

A number of circulating miRNAs were found significantly associated with LV hypertrophy and fibrosis [8,21,22]. In this study, we further evaluated the relationship between myocardial miRNAs and LVH, myocardial fibrosis and LV remodeling.

Previous studies focused on circulating miR-27a, miR-29a and miR-199a-5p, which were validated to be significantly correlated with LVH and LV fibrosis in various studies [8,22]. Our study found that myocardial hsa-miR-3960 and hsa-miR-652-3p were significantly associated with LV hypertrophy, and hsa-miR-642a-3p was significantly correlated with LV fibrosis; this means that myocardial miRNAs could also perform as biomarkers of LVH and LV fibrosis. In addition, we also found that hsa-miR-499a-5p, hsa-miR-7107-5p, hsa-miR-3141 and hsa-miR-3679-5p were significantly associated with the remodeling of the LV cavity. To our knowledge, this has not been studied yet.

MiR-499 was shown to regulate the cardiac β-MyHC/α-MyHC ratio, and β-MyHC expression was increased in human hearts of patients with ischemic cardiomyopathy and other heart diseases [23,24,25,26]. Circulating miR-499a-5p was proven as one of the dysregulated miRNAs in HCM, which expressed higher in HCM patients than in healthy controls, and carriers of P/LP variants in the MYH7 gene expressed higher levels than in controls [27,28]. In our study, we found that myocardial miR-499a-5p was also up-regulated, and higher expression of myocardial miR-499a-5p correlated with lower LVEF. However, the relationship between LV remodeling and miR-499a-5p needs further validation and mechanism exploration.

MiR-642-3p was revealed as an adipocyte-specific microRNA in a previous study [29], but the function of miR-642 families on myocytes has not been reported. So, although myocardial miR-642-3p was significantly down-regulated in hypertrophied myocardium and correlated with LV fibrosis in our study, further studies are needed to explore the mechanism of this relationship.

In the study by Eyyupkoca et al. [30], mir-652-3p was identified to be associated with adverse left ventricular remodeling (ALVR), which was defined as an increase in LVEDV and LVESV > 13% 6 months after acute myocardial infarction (AMI). Unfortunately, studies were limited in other miRNAs, such as miR-3960, miR-7107-5p, miR-3141 and miR-3679-5p in the cardiovascular field, so greater efforts are needed in this field to explore the mechanism of how these miRNAs work and to find potential therapeutic targets.

Several studies have suggested that endogenic factors such as myeloperoxidase (MPO) and nitric oxide synthase (NOS) may play important roles in the frail population [31,32]. However, the effect of endogenic factors derived from myocardial tissue on HCM patients was not investigated in this study, and further study is needed to explore the possible protective role or the unfavorable role of endogenic factors in the HCM population.

There are several limitations to our study. First, since the myocardial species were difficult to obtain, especially in patients without HCM, we only included 21 HCM patients and four healthy controls. So, our result needs further validation with large-scale studies. Then, due to insufficient image quality, T1 maps were not available for analysis in many patients, so we did not analyze the relationship between myocardial miRNAs and diffuse myocardial fibrosis. Next, we did not collect blood samples from these patients, so it is impossible to analyze the relationship between circulating miRNAs and myocardial miRNAs; this will be improved in future research. Finally, patients’ follow-up was not conducted in this study, and the relationship between myocardial miRNAs and long-term outcomes was not discussed; this will be added in the next research.

## 5. Conclusions

In conclusion, our study found that 68 myocardial miRNAs were significantly increased or decreased in the HCM group. Myocardial miRNA levels could be used as potential biomarkers for LV hypertrophy, fibrosis and remodeling.

## Figures and Tables

**Figure 1 diagnostics-12-01978-f001:**
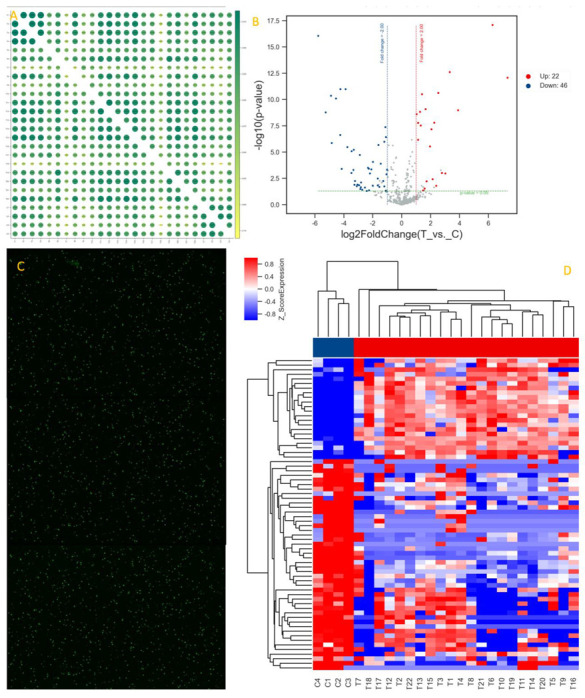
MiRNAs extraction and screening process. (**A**) Correlation of gene expression levels among samples using Pearson correlation methods. (**B**) Volcano map, in which each point represents a miRNA, the abscissa represents the logarithm of the difference multiple of the expression of a certain miRNA in the two groups of samples (log2fc), and the ordinate represents the negative logarithm (−log10 (*p*-value)) of the statistical significance of the change of miRNA expression. The greater the absolute value of abscissa, the greater the difference multiple between the two groups; The larger the ordinate is, the more significant the differential expression is, and the more reliable the differentially expressed miRNA screened is. Red dots indicate up-regulation, blue dots indicate down-regulation, and gray dots indicate that at least 75% of the samples in one group are marked as ‘ detected ‘ or non-differentially expressed miRNA. (**C**) Raw data scanning diagram of a chip, each point on the diagram corresponds to a probe (only shows a partial probe). (**D**) Clustering relationship between samples, which could distinguish two or more groups of samples.

**Figure 2 diagnostics-12-01978-f002:**
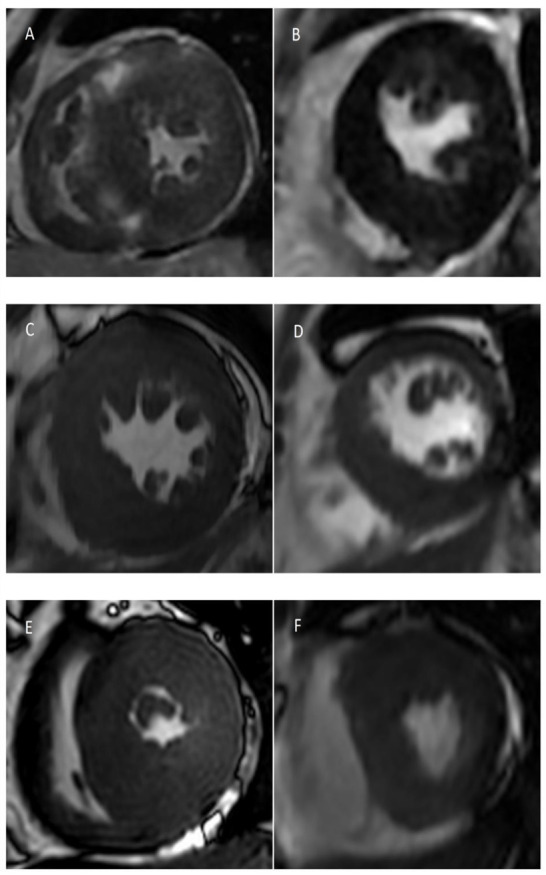
CMR findings in different patients with different myocardial miRNA levels. (**A**) Short axis LGE image in a patient with high expression of myocardial hsa-miR-642a-3p showed extensive LGE in the septal wall of LV. (**B**) Short axis LGE image in a patient with low expression of myocardial hsa-miR-642a-3p, no LGE was found in the LV myocardial. (**C**) Short axis cine image in a patient with high expression of myocardial hsa-miR-652-3p and hsa-miR-3960 showed significantly thickened myocardial. (**D**) Short axis cine image in a patient with low expression of myocardial hsa-miR-652-3p and hsa-miR-3960 showed myocardial thickening but was not as obvious as the previous one. (**E**) Short axis cine image in a patient with high expression of myocardial hsa-miR-3679-5p and hsa-miR-3141 showed low LVESVi. (**F**) Short axis cine image in a patient with low expression of myocardial hsa-miR-3679-5p and hsa-miR-3141 showed higher LVESVi than the previous one. **CMR**, cardiovascular magnetic resonance; **LGE**, late gadolinium enhancement; **LV**, left ventricle; **LVESVi**, left ventricular end systolic volume index.

**Figure 3 diagnostics-12-01978-f003:**
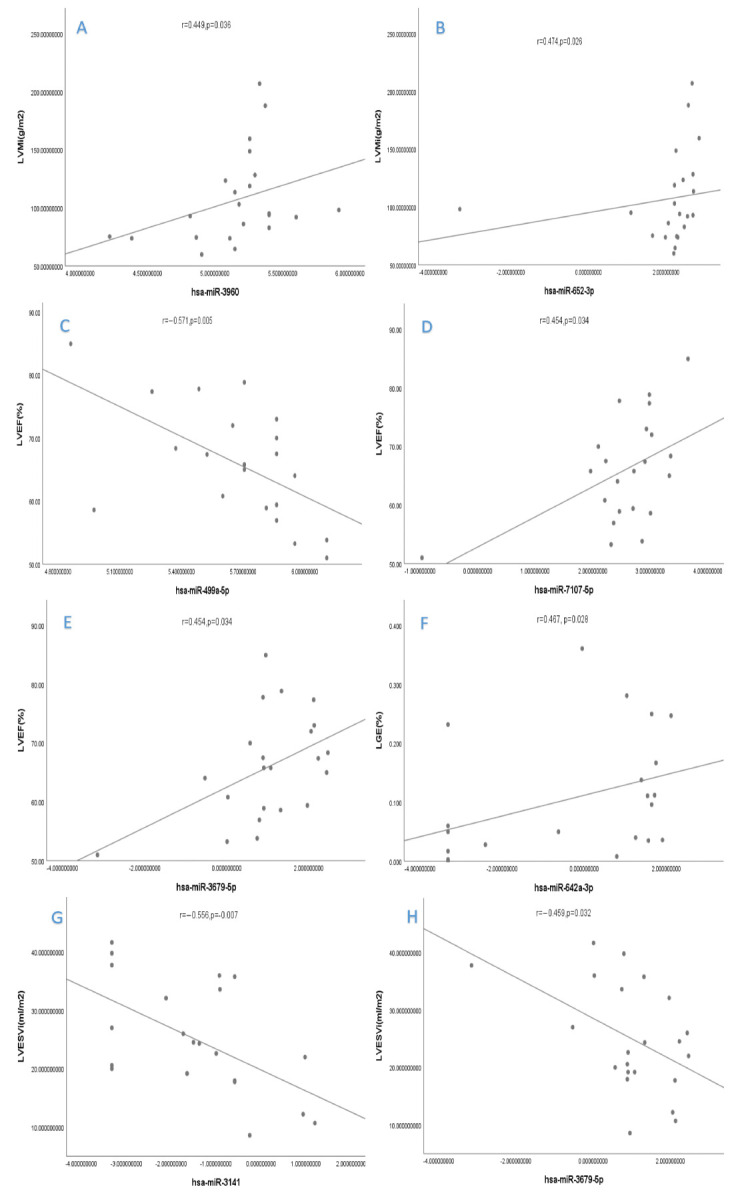
Correlation between miRNA levels and clinical variables. (**A**) The expression levels of hsa−miR−3960 were significantly and positively correlated with LVMi. (**B**) The expression levels of hsa−miR−652−3p were significantly and positively correlated with LVMi. (**C**) The expression lev of hsa−miR−499a−5p was significantly and negatively correlated with LVEF. (**D**) The expression levels of hsa−miR−7107−5p were significantly and positively correlated with LVEF. (**E**) The expression levels of hsa−miR−3679−5p were significantly and positively correlated with LVEF. (**F**) Hsa−miR−642a−3p expression was positively correlated to the quantification of LGE. (**G**) Hsa−miR−3141 expression was negatively correlated to LVESVi. (**H**) Hsa−miR−3679−5p expression was negatively correlated to LVESVi. **LVMi**, left ventricular mass index; **LVEF**, left ventricular ejection fraction; **LGE**, late gadolinium enhancement; **LVESVi**, left ventricular end systolic volume index.

**Table 1 diagnostics-12-01978-t001:** Clinical characteristics of HCM patients and healthy controls.

	HCM Patients		Controls		*p* Value
	*n*	Value	*n*	Value	
Age (years)	21	55 (41,60)	4	51 (32,59)	0.738
Male	8	38%	2	50%	1.000
Significant LVOT gradient (>30 mm Hg)	19	90%	0	0%	-
Family history of HCM	4	19%	0	0%	-
History of syncope	5	24%	0	0%	-
Nonsustained tachycardia	1	5%	0	0%	-
Maximal EDWT > 30 mm	2	10%	0	0%	-
Mitral regurgitation			0	0%	-
Mild	2	10%	0	0%	-
Moderate	7	33%	0	0%	-
Severe	12	57%	0	0%	-

HCM, hypertrophic cardiomyopathy; LVOT, left ventricular outflow tract; EDWT, end diastolic left ventricular wall thickness. Quantitative data were expressed as the median and interquartile range (IQR).

**Table 2 diagnostics-12-01978-t002:** CMR and echocardiography findings of HCM patients.

CMR Parameters	HCM Patients
Left atrial volume index (mL/m^2^)	53 (44,68)
Left ventricular ejection fraction (%)	66 (59,73)
Left ventricular end diastolic volume index (mm/m^2^)	75 (57,88)
Left ventricular end systolic volume index (mm/m^2^)	24 (19,35)
Left ventricular outflow tract gradient (mm Hg)	78 (54,101)
Maximal wall thickness (mm)	22 (18,28)
Left ventricular mass index (mm/m^2^)	95 (79,126)
Quantification of late gadolinium enhancement (%)	6 (2,20)

CMR, cardiac magnetic resonance; HCM, hypertrophic cardiomyopathy. Quantitative data were expressed as the median and interquartile range (IQR).

**Table 3 diagnostics-12-01978-t003:** Up-regulated and down-regulated miRNAs in HCM patients.

miRNAs	*p*-Value	FoldChange	log2FoldChange	Regulation
hsa-miR-15a-5p	1.67 × 10^−8^	2.190249138	1.131094984	Up
hsa-miR-24-1-5p	6.63 × 10^−7^	2.185726725	1.128113036	Up
hsa-miR-95-3p	0.00107674	8.082612468	3.014821677	Up
hsa-miR-208a-3p	1.37 × 10^−6^	−28.85041914	−4.850520374	Down
hsa-miR-148a-3p	0.007665064	−7.04991722	−2.817606318	Down
hsa-miR-10a-5p	0.023519485	−5.823609987	−2.54191374	Down
hsa-miR-181a-5p	2.42 × 10^−9^	2.046487301	1.033149714	Up
hsa-miR-181b-5p	1.61 × 10^−8^	4.767522463	2.253239736	Up
hsa-miR-204-5p	0.000980838	−11.19625089	−3.484943816	Down
hsa-miR-218-5p	8.97 × 10^−17^	−54.93956941	−5.779773698	Down
hsa-miR-221-3p	8.08 × 10^−18^	77.96215989	6.284702155	Up
hsa-miR-142-3p	0.00037835	−4.857959151	−2.280350359	Down
hsa-miR-149-5p	2.81 × 10^−6^	3.888492702	1.959211031	Up
hsa-miR-188-5p	0.001074418	−4.594043979	−2.199764668	Down
hsa-miR-378a-5p	2.43 × 10^−13^	9.967233981	3.317193196	Up
hsa-miR-378a-3p	1.45 × 10^−9^	2.409517431	1.268744238	Up
hsa-miR-424-5p	0.015718634	5.24821437	2.39182665	Up
hsa-miR-451a	0.048520532	−2.158469131	−1.110008461	Down
hsa-miR-486-5p	3.01 × 10^−8^	2.491376359	1.316942978	Up
hsa-miR-499a-5p	8.04 × 10^−10^	3.125868037	1.644256874	Up
hsa-miR-551b-3p	7.91 × 10^−11^	−23.50718535	−4.555029902	Down
hsa-miR-575	1.02 × 10^−11^	−18.94850888	−4.244012418	Down
hsa-miR-652-3p	0.003807107	4.396032747	2.136202133	Up
hsa-miR-23b-5p	0.006018149	3.310903689	1.727225045	Up
hsa-miR-125a-3p	0.003260678	−3.153527041	−1.656966304	Down
hsa-miR-486-3p	0.000950208	6.877931643	2.781974778	Up
hsa-miR-490-5p	0.005608572	−2.417505315	−1.273519062	Down
hsa-miR-455-3p	6.87 × 10^−8^	4.151633047	2.053678933	Up
hsa-miR-208b-3p	8.33 × 10^−13^	159.1822033	7.31453524	Up
hsa-miR-1225-5p	1.02 × 10^−6^	−2.301696596	−1.202697674	Down
hsa-miR-1246	1.03 × 10^−11^	−14.79899608	−3.887427406	Down
hsa-miR-1915-3p	0.012200862	−8.341626335	−3.060328688	Down
hsa-miR-224-3p	1.04 × 10^−9^	14.97603971	3.904584261	Up
hsa-miR-3141	0.015232493	−4.006507145	−2.00234505	Down
hsa-miR-4298	0.042552949	−4.810758055	−2.266264244	Down
hsa-miR-4270	0.012522098	−9.686961125	−3.276044152	Down
hsa-miR-4291	0.029289007	2.996508744	1.583282584	Up
hsa-miR-3679-5p	0.014999028	−3.151810102	−1.656180614	Down
hsa-miR-378d	2.36 × 10^−11^	5.814021124	2.539536313	Up
hsa-miR-4442	0.014525527	−7.714730539	−2.947615767	Down
hsa-miR-4466	0.000397024	−17.79445123	−4.153355537	Down
hsa-miR-4530	3.77 × 10^−7^	−2.141285992	−1.098477496	Down
hsa-miR-378i	3.05 × 10^−11^	2.634035175	1.397274612	Up
hsa-miR-3960	0.000482361	−2.028841036	−1.020655831	Down
hsa-miR-4634	1.85 × 10^−5^	−7.491169168	−2.905190902	Down
hsa-miR-4669	0.032262486	−7.26027626	−2.860024445	Down
hsa-miR-4687-3p	2.06 × 10^−7^	−3.042668733	−1.605337271	Down
hsa-miR-4741	1.67 × 10^−9^	−38.44059936	−5.264558926	Down
hsa-miR-4787-5p	0.000435723	−4.295676251	−2.102885267	Down
hsa-miR-4788	2.32 × 10^−7^	−19.06965697	−4.253206987	Down
hsa-miR-642a-3p	0.049097861	−5.377734753	−2.426998599	Down
hsa-miR-5001-5p	8.45 × 10^−5^	−3.991371678	−1.996884629	Down
hsa-miR-1229-5p	0.005670955	−9.395082827	−3.231905881	Down
hsa-miR-5787	4.37 × 10^−11^	−29.79856364	−4.897170886	Down
hsa-miR-6088	0.00125777	−2.075538432	−1.053485646	Down
hsa-miR-6090	0.000126781	−2.72669402	−1.447152815	Down
hsa-miR-6510-5p	0.020496139	−2.921621676	−1.546769374	Down
hsa-miR-208a-5p	8.26 × 10^−6^	−11.036198	−3.464171341	Down
hsa-miR-6727-5p	0.000316337	−5.062208252	−2.339766859	Down
hsa-miR-6739-5p	0.044423071	2.783974922	1.477146215	Up
hsa-miR-6800-5p	0.017351182	−8.6565924	−3.113799231	Down
hsa-miR-6850-5p	0.000487495	−10.15242969	−3.343753131	Down
hsa-miR-6891-5p	0.021973529	−7.101027763	−2.828027847	Down
hsa-miR-7107-5p	0.017734323	−2.296376215	−1.199359018	Down
hsa-miR-7110-5p	0.037958046	−6.662449596	−2.736052713	Down
hsa-miR-7150	3.40 × 10^−6^	−13.56395008	−3.761705475	Down
hsa-miR-7847-3p	6.05 × 10^−6^	−10.01635556	−3.324285775	Down
hsa-miR-8069	4.29 × 10^−8^	−2.194231964	−1.133716049	Down

HCM, hypertrophic cardiomyopathy.

## Data Availability

The data presented in this study are available on request from the corresponding author.

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
