# Peer review of "Differential Expression of microRNAs in Hypertrophied Myocardium and Their Relationship to Late Gadolinium Enhancement, Left Ventricular Hypertrophy and Remodeling in Hypertrophic Cardiomyopathy"

_diagnostics, 2022, doi:10.3390/diagnostics12081978_

Round 1

Reviewer 1 Report

The authors analyzed differential expression of micro RNAs among HCM patients with correlation to CMR parameters. The aim of the paper shows that myocardial miRNA level could be used as a biomarkers and become a promising tool in diagnosis of fibrosis, remodeling and LV hypertrophy. 

The main potential contributions of the paper are threefold. First, the authors introduced methodology and study population – clinical characteristic.

Second, the authors reported results - significantly increased expression of 22 miRNAs in HCM group, while 46 miRNAs were noticed as a significantly decreased level in HCM group compared to control group. Third, the correlation between miRNA and CMR parameters such as (LVMi, LVEF, LVESVi , LGE) were also analyzed . 

Introduction leads us from general subject area to a particular topic of inquiry. It establishes significance of the research such as background informations and aim of the paper.

The materials and methods section introduce the detailed methodological approach. Results section consist of patients characteristics ( clinical characteristics , CMR findings ), up-regulated and down-regulated miRNAs and correlation with CMR parameters.  

The discussion describes the significance of findings compared to previous research. Several limitations were also included in study.

I find this article very interesting and relevant. The study should be continued in the future. It will be interesting to find results with large scale studies, analysis relationship between myocardial miRNAs and myocardial fibrosis and LV remodeling likewise.

Suggestions : Clinical characteristics of patients should be in materials and methods section.

Author Response

Thanks for your professional and constructive comment for our article. We have carefully considered your suggestion and made changes. We have tried our best to improve and made changes in the manuscript.

1.”Suggestions : Clinical characteristics of patients should be in materials and methods section.”

Reply:We noted that clinical characteristics of patients should be list in materials and methods section and made correction(see in page 2-3, line 86-93, table 1), thank you for putting forward this question.

2.”I find this article very interesting and relevant. The study should be continued in the future. It will be interesting to find results with large scale studies, analysis relationship between myocardial miRNAs and myocardial fibrosis and LV remodeling likewise.”

Reply:This study will be continued in the future with a larger scale to further analysis the relationship between myocardial miRNAs and myocardial fibrosis and LV remodeling. Thanks for your concern for our study.

Finally, Thank you again for taking your time to review this manuscript. We really appreciate all your comments and suggestions!

Reviewer 2 Report

Authors present a very important and interesting study well written and discussed. It is suitable for publication. 

The paper proposed  myocardial microRNA as a prognostic factor to better customize treatment and therapy. The clinical value of the paper is high, it is a promising topic. The study is rigorous and well written.

I just suggest the possible interaction and protective role (10.1007/s11357-012-9428-4) and the unfavorable role (10.1093/gerona/glp183) of endogenic factors in frail population.

Author Response

Thanks for your professional and constructive comment for our article. We have carefully considered your suggestion and made changes. We have tried our best to improve and made changes in the manuscript.

1.”I just suggest the possible interaction and protective role (10.1007/s11357-012-9428-4) and the unfavorable role (10.1093/gerona/glp183) of endogenic factors in frail population.”

Reply: Thanks for your suggestion. We have added this part in discussion section and improved cited references(see in page 12-13, line 283-289, reference 31-32).

Finally, Thank you again for taking your time to review this manuscript. We really appreciate all your comments and suggestions!